Review Article

# Type IV secretion systems: from structures to mechanisms

Pierre Paillard [ID][1,2], Quentin Rouger[1,2], Manon Thomet [ID][1] & Kévin Macé [ID][1✉]

## Abstract

Bacterial conjugation is the fundamental process of unidirectional transfer of DNA from a "donor" cell to a "recipient" cell. It is the primary means by which antibiotic resistance genes spread among bacterial populations. Conjugation is mediated by a large molecular machinery termed Type IV secretion system (T4SS), embedded within the donor cell wall. In addition, some bacteria utilise T4SS to inject effector proteins into eukaryotic cells, modulating host functions to their advantage. In this review, we highlight how recent structural studies have substantially advanced our understanding of T4SS molecular mechanisms. We detail these mechanisms across four main sub-processes: assembly of the machinery, pilus biogenesis, donor–recipient cell contact, and substrate recruitment and secretion. By understanding the intricate workings of T4SS, we can gain valuable insights into bacterial evolution, virulence, and horizontal gene transfer, offering potential avenues for developing novel antibacterial strategies.

**Keywords** Type IV Secretion System; T4SS; Secretion; Conjugation; Molecular Mechanisms
**Subject Categories** Microbiology, Virology & Host Pathogen Interaction; Structural Biology

## Introduction

Type IV secretion systems (T4SS) are complex multiprotein machineries found in many bacteria, with the unique ability to translocate both DNA and protein substrates into bacterial or eukaryotic recipient cells (Cascales and Christie, 2003). T4SSs display a wide range of biological activities, including a critical role in horizontal gene transfer (HGT) and the establishment of bacteria-eukaryotic host interactions. The transfer of DNA between bacteria, known as conjugation, is the primary mechanism for horizontal gene transfer (Botelho and Schulenburg, 2021). As bacteria reproduce asexually, conjugation plays a critical role in bacterial evolution. Historically, this process was referred to as "sex pili", "sex factor", or "conjugal fertility", with the donor cell likened to the "male" and the recipient to the "female", highlighting its role

in facilitating genetic exchange between cells (Lawn et al, 1967; Redfield, 2001). Nowadays, conjugation has garnered attention as the principal route for the dissemination of antibiotic resistance genes, a main contributor to the global antibiotic resistance crisis (Holmes et al, 2016). Beyond conjugation, bacteria use T4SS to interact with eukaryotic cells by injecting effector proteins, which can either promote symbiotic relationships or drive pathogenic processes (Celli and Tsolis, 2015; Elhenawy et al, 2021). These proteins modulate host cell functions to the bacteria's advantage, enabling pathogenic bacteria to evade the immune system and establish infections (Celli and Tsolis, 2015). For example, *Legionella pneumophila*, the causative agent of Legionnaires' disease, and *Helicobacter pylori*, linked to gastric ulcers and cancer, both employ T4SS to manipulate host cells and promote bacterial survival (Qiu and Luo, 2017; Kwok et al, 2007).

The dual functionality of T4SS, i.e., for both DNA transfer in bacterial conjugation and protein injection in host-pathogen interactions, makes it a pivotal system in bacterial evolution and virulence. Since its discovery in the mid-20th century (Lederberg and Tatum, 1946), research on T4SS has led to significant advancements in molecular genetics and biotechnology. For example, T4SS has been used for gene mapping of *Escherichia coli*, opening the field of molecular genetics. Later, *Agrobacterium tumefaciens* was used to create the first genetically modified plants through its T4SS (Fraley et al, 1983), marking a breakthrough in genetic engineering (Giddings et al, 2000). Looking ahead, research on T4SS holds significant potential for innovative applications (Heggie et al, 2024). For instance, phage therapies that exploit filamentous bacteriophages binding specifically to conjugative T4SS could help combat antibiotic-resistant bacteria (Meng et al, 2019; Conners et al, 2023). In addition, engineered T4S systems are being explored for delivering therapeutic agents, such as CRISPR-Cas9, into specific cells, thus opening up new possibilities for precision medicine (Guzmán-Herrador et al, 2024).

Despite extensive research, the molecular mechanisms of T4SS function had long remained elusive, largely due to the complexity of its structure and its diverse biological functions. However, recent breakthroughs in structural biology have provided unprecedented insights into the architecture and function of T4SS. This review focuses on these new structural discoveries and their significant contributions to our understanding of molecular mechanisms underlying T4SS assembly and function. Specifically, we will explore four chronological key sub-processes: machinery assembly, pilus biogenesis, donor–recipient cell contact, and substrate

[1]Univ. Rennes, CNRS, Institut de Génétique et Développement de Rennes (IGDR) - UMR6290, 35000 Rennes, France. [2]These authors contributed equally: Pierre Paillard, Quentin Rouger. ✉E-mail: kevin.mace@univ-rennes.fr

recruitment–secretion. Understanding these mechanisms is essential not only for advancing fundamental biological knowledge but also for developing innovative therapeutic strategies to combat antibiotic resistance, pathogenicity, and potentially revolutionising future medicine.

# T4SS composition and structural organisation

The nomenclature of T4SS is complex due to their broad functional diversity and presence across various bacterial species. Naming conventions vary based on system size, bacterial phyla, substrate type (DNA or proteins), and whether they are plasmid-encoded or chromosomally integrated. In addition, the low sequence conservation among T4SS proteins makes the identification of homologues across different systems challenging and sometimes misleading. To address these inconsistencies and ensure accurate protein classification, we have compiled a comprehensive reference table, integrating data from UniProt and AlphaFold databases (Table 1).

## Protein by protein: structure/function

T4SS comprises several essential proteins, each playing a specific role within the machinery. VirB1 contains a transglycosylase domain, which degrades peptidoglycan to create an opening hole in the bacterial wall, allowing for T4SS assembly (Koraimann, 2003). VirB2, a pilin initially located in the inner membrane (IM), is extracted along with a phospholipid to form a pilus, which mediates surface contact between donor and recipient cells, and establishes a channel for substrate transfer (Hospenthal et al, 2017). VirB3 integrates into the IM with two transmembrane helices and interacts strongly with VirB4—sometimes both are fused into a single protein (Batchelor et al, 2004; Mossey et al, 2010). VirB4 is a large cytoplasmic ATPase that supplies energy for pilus biogenesis. VirB4 forms a hexamer of dimers, interacting with VirB8, VirB10, and VirB11 proteins (Chetrit et al, 2018). VirB5 forms the pilus tip that is involved in recipient cell recognition and potentially in pore formation (Aly and Baron, 2007). Before the initiation of the pilus assembly, VirB5 is located in the periplasm on top of VirB6, which recruits VirB2 subunits to support pilus assembly (Macé et al, 2022b). VirB7, a lipoprotein, stabilises the outer membrane core complex (OMCC) and anchors it to the outer membrane (OM) (Oliveira et al, 2016). VirB8 participates in pilus assembly through its periplasmic domain and interacts with VirB4 *via* its tail domain, contributing to the anchoring of the inner membrane complex (IMC) into the IM (Macé et al, 2022b). VirB9, alongside with the C-terminal domain of VirB10, forms part of the OMCC, which guides the pilus and in a closed state blocks entrance of extracellular material (Chandran et al, 2009). VirB10, which spans from the cytoplasm to the outer membrane, is the key component of the machinery. By interacting with almost all proteins, VirB10 maintains structural integrity of the entire T4SS machinery and has multiple additional functions, such as regulation of pilus biogenesis or ATPase activity sensing (Fronzes et al, 2009). VirB11 is a cytoplasmic ATPase that works in tandem with VirB4 to energise pilus formation (Chetrit et al, 2018). Finally, VirD4 is another cytoplasmic ATPase that is anchored to the inner membrane through multiple transmembrane segments, which form a channel

across the membrane and function to recruit and translocate substrates into the T4SS (Tato et al, 2005). In addition, for delivery of DNA substrates, at least one additional protein is required: the relaxase, which recognises and nicks the OriT site on the DNA. Covalently bound to the DNA, the relaxosome complex is then recruited and secreted by the T4SS into the recipient cell. Relaxases are generally the largest and most variable proteins in conjugative T4S systems (Ilangovan et al, 2015).

## Global structural organisation

Recent cryo-electron tomography and single-particle cryo-electron microscopy (cryo-EM) studies have revealed that, except in Gram-positive bacteria and some rare exceptions among Gram-negative bacteria, T4SS machineries are composed of at least 12 core components, which assemble into four distinct subcomplexes. T4SSs built from these 12 core components (VirB1 to VirB11 and VirD4) are referred to as "minimised" systems (e.g, the $T4SS_{R388}$ prototype). In contrast, "expanded" T4SSs (e.g, the $T4SS_{F-plasmid}$ prototype) contain additional protein subunits that may assemble into additional subcomplexes (Chetrit et al, 2018). This structural evolution, i.e., the extension from an ancestral core complex, is similar to that observed in other biological machineries, such as the ribosome (Anger et al, 2013).

The four core subcomplexes are: the Outer Membrane Core Complex (OMCC), the Stalk, the Arches, and the Inner Membrane Complex (IMC) (Macé et al, 2022b; Low et al, 2014) (Fig. 1A). The OMCC, composed of VirB7, VirB9, and VirB10, exhibits an unusual symmetry mismatch between the O-layer embedded in the outer membrane and the I-layer beneath it (Macé and Waksman, 2024; Amin et al, 2021; Durie et al, 2020; Sheedlo et al, 2020) (Fig. 1A). The Stalk is a fivefold symmetrical structure consisting of two main components: VirB5, positioned at the top of the Stalk before being recruited as pilus tip, and VirB6, embedded in the IM and hypothesised to serve as a platform for pilus subunit recruitment and assembly (Macé et al, 2022b) (Fig. 1B). Surrounding the Stalk are the Arches, forming a sixfold symmetrical ring. The Arches is made of six tetramers of the periplasmic domain of VirB8, along with two small regions of VirB10 that span from the OMCC and extend downwards to interact with the IMC (Macé and Waksman, 2024) (Fig. 1B). The IMC consists of six copies of a protomer comprising VirB3, two subunits of VirB4 ($VirB4_{central}$ and $VirB4_{outside}$), and the N-terminal tails of four VirB8 molecules. $VirB4_{central}$ forms a central hexamer linked to the IM through VirB3, while $VirB4_{outside}$ forms a dimer with $VirB4_{central}$ and interacts with the transmembrane tails of VirB8 (Macé et al, 2022b). VirB10, originating from the Arches, crosses the inner membrane by interacting with VirB6, and then extends into the cytoplasm to interact with both central and outside VirB4 and VirB3 (Macé and Waksman, 2024) (Fig. 1B). VirB10 can be subdivided into distinct sub-domains, from the outer membrane to the cytoplasm: $VirB10_{O-layer}$, $VirB10_{I-Layer}$, $VirB10_{Arches}$, $VirB10_{IM}$ and $VirB10_{Cyto}$ (Macé and Waksman, 2024). The two lower domains, $VirB10_{IM}$ and $VirB10_{Cyto}$, also interact with VirD4, although the precise location of VirD4 within the T4SS machinery remains elusive. The final IMC component, VirB11, is positioned beneath the $VirB4_{central}$ hexamer (Chetrit et al, 2018), forming a double-ring ATPase complex to generate the energy required for pilus biogenesis (Hospenthal et al, 2017).

**Table 1. Nomenclature and functions of core T4SS proteins across reference models.**

| Name / Group / Strains | Ti-plasmid / – / A.t. | R388 / IncW / Ent.b | F-plasmid / IncF / Ent.b | RP4 / IncP / Proto.b | R64 / IncI / Entero.b | pCF10 / – / G + E.f. | ICEST3 / – / G + S.t. | X-T4SS / – / X.c. | Cag / – / H.p. | dot/icm / – / L.p. | Subcomplex | Localisation | nbr of copies (R388) |
|---|---|---|---|---|---|---|---|---|---|---|---|---|---|
| **Proteins** | | | | | | | | | | | | | |
| VirB1 P0A3V6 | TrwN Q6I6C7 | Orf19 P47737 | TrbN Q03538 | – | PrgK Q5G3P4 | OrfA Q70C98 | VirB1 A0A0U5FAS56 | Cagγ/Cag4 Q75X65 | – | – | Peri | Dimer |
| VirB2 P09776 | TrwL O50328 | TraA P04737 | TrbC Q79DP1 | TraR Q9WW98 | – | – | TrwL A0A0U5FEF2 | CagC Q75X97 | – | Pilus | IM/extra | Polymer |
| VirB3 P0A3V8 | TrwM O50329 | TraL P08321 | TrbD Q79DP0 | traK Q9WW99 | PrgI Q5G3P6 | OrfE Q70CA2 | VirB3 A0A0U3A649 | CagE Q48252 | IcmT Q6QFI8 | IMC | IM/cyto | 6 monomers |
| VirB4 P0A3W0 | TrwK O50330 | TraC P18004 | TrbE Q52363 | TraU Q9R2G9 | PrgJ Q5G3P5 | OrfD Q70CA1 | VirB4 A0A0U5FG66 | | DotO/IcmB Q6QG55 | IMC | cyto | 12 (hexamer of dimer) |
| VirB5 P0A3W2 | TrwJ O50331 | – | TrbJ Q03534 | TraW Q9WW97 | – | – | VirB5 A0A0U3BHC9 | CagL Q75X76 | IcmX Q6QFE2 | Stalk/Pilus | peri | Pentamer |
| VirB6 P09779 | TrwI O50333 | TraG P33790 | TrbL Q03536 | TraY Q9WW95 | PrgH Q5G3P7 | OrfC Q70CA0 | TrwI A0A0U5FDT2 | – | DotA Q6Y092 | Stalk | IM/peri | Pentamer |
| VirB7 P0A3W4 | TrwH O50334 | TraV P41069 | TrbH Q03542 | – | – | – | VirB7 A0A0U5FGZ7 | CagT P97245 | DotD O52183 | OMCC | peri/OM | 14 monomers |
| VirB8 P09781 | TrwG O50335 | TraE P08322 | TrbF Q03540 | TraM Q9R2H2 | PrgL Q5G3P3 | OrfG Q70CA4 | TrwG A0A0U5FGD7 | CagV Q75X30 | DotI Q7ATT8 | Arches/IMC | IM/peri | 24 (6×4) |
| VirB9 P0A3W6 | TrwF O50336 | TraK P41066 | TrbG Q03541 | TraN Q9R2H1 | – | – | TrwF A0A0U5FAB7 | CagX Q75X32 | DotH O54589 | OMCC | peri | 14/16 |
| VirB10 P09783 | TrwE O50337 | TraB P41067 | TrbI Q03543 | TraO Q9R2H0 | – | – | TrwE A0A0U5FAA5 | CagY A0A4Y4W4F3 | DotG/IcmE Q5WZ89 | OMCC/Arches/IMC | OM/peri/IM/cyto | 14/16/12/2×6 |
| VirB11 P0A3F8 | TrwD O50338 | – | TrbB Q79DP2 | TraJ Q7DJM3 | – | – | TrwD A0A0U5F9J8 | Cagα/VirB11 Q75XE5 | DotB Q5ZS43 | IMC | cyto | Hexamer |
| VirD4 Q44360 | TrwB Q04230 | TraD P09130 | TraG Q00185 | TrbC Q56I02 | PcfC Q5G3N9 | OrfK Q70CA8 | TrwB A0A0U5FBR8 | Cagβ/Cag5 Q75X64 | DotL/IcmO Q5ZYC6 | IMC | cyto/IM | Hexamer |

A.t. *Agrobacterium tumefaciens*, Entero.b *enterobacteria*, Proto.b *proteobacteria* (alpha, beta and gamma) (Soda et al, 2008), G + E.f. Gram-positive *Enteroccous faecalis*, G + S.t. Gram-positive *Streptococcus thermophilus*, X.c. *Xanthomonas citri*, H.p. *Helicobacter pylori*, L.p. *Legionella pneumophila*.

The table summarises the nomenclature and functions of the 12 core proteins of Type IV Secretion Systems (T4SS) across reference models, indicating the bacterial strain of origin, the exclusion group when available, and the corresponding UniProt ID in italics. Protein classification is based on literature-supported homology (Virolle et al, 2020; Breidenstein et al, 2025) and refined using AlphaFold structural predictions. For *Agrobacterium tumefaciens*, bold font is used to highlight the consensus nomenclature of the proteins.

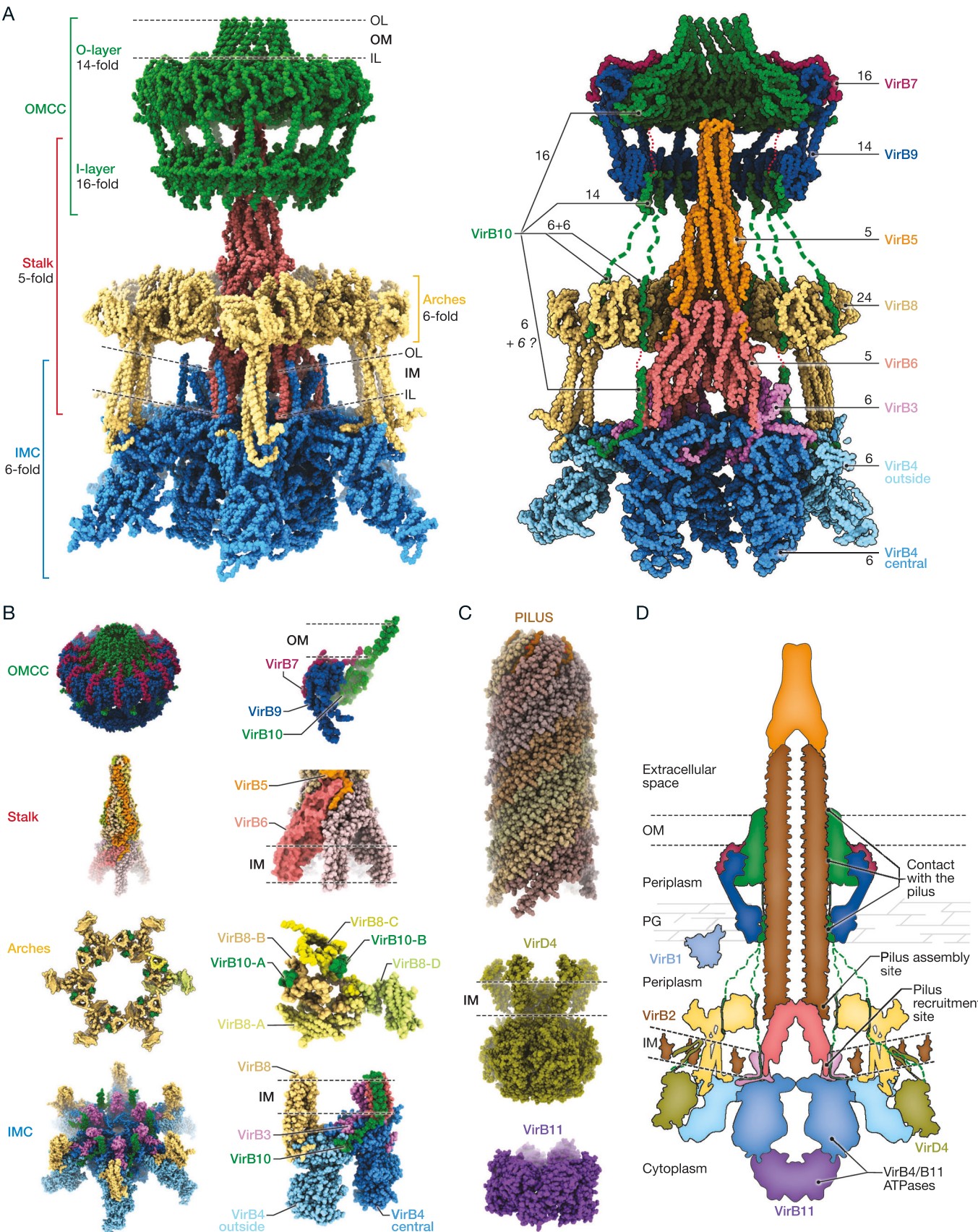

**Figure 1. Global structural organisation of T4SS.**

(A) Left, Assembled structure of T4SS$_{R388}$ in surface representation, without side-chain for clarity, and coloured by subcomplex ((Protein Data Bank (PDB) identifier (ID) 7O3J, PDB ID 7O3T, PDB ID 7O3V, PDB ID 7Q1V and PDB ID 7OIU). Right, Cut-view of the assembled structure of T4SS$_{R388}$, coloured by individual proteins. Membrane localisation and the oligomerisation state of the protein are indicated. (B) Subcomplex organisation and protomers. Zoomed-in view of each T4SS subcomplex, represented in surface with side-chains shown and coloured by protein. Black dashed lines indicate the boundaries of the outer membrane and inner membrane. (C) Separately solved structures of the T4SS assembly. Same representation as in (B). Top, Pilus R388 (PDB ID 8S6H). Middle, VirD4$_{TrwB}$ R388 (Alphafold3 and PDB ID 1GKI). Bottom, VirB11$_{Caga}$ from *Helicobacter pylori* (PDB ID 1NLZ). (D) Cut-view schematic representation of T4SS during pilus biogenesis. The cellular compartments and key locations are indicated, with colour-coding consistent with previous panels.

The T4SS structural organisation described above corresponds to the resting-state of the T4SS. However, the system likely undergoes dynamic rearrangements depending on its activities. Following assembly, the T4SS constructs an extracellular filament known as a pilus, which facilitates contact between the donor and the recipient cell. Based on structural data (Fig. 1C), a plausible model of the T4SS during pilus biogenesis has been proposed (Fig. 1D) (Macé et al, 2022b). In this model, the IMC forms a large intramembrane empty space, analogous to a "station hall", optimising the arrival of VirB2 pilin subunits at the central assembly sites on VirB6. Pilus assembly proceeds in a stepwise fashion, with layers of VirB2 added sequentially (Hospenthal et al, 2017). The growing pilus then passes through the OMCC, where the flexibility provided by the mismatch between the O-layer and I-layer facilitates pilus accommodation and biogenesis (Costa et al, 2024). In addition, the OMCC acts as a guide, ensuring that the pilus traverses the outer membrane in a perpendicular way, while stabilising the pilus attachment to the rest of the T4SS (Hu et al, 2019). This stabilisation is relevant, since high forces could be applied on the pilus during contact with the recipient cell (Clarke et al, 2008). The Arches, located around the base of the pilus (Fig. 1D), likely reinforce the pilus attachment to the machinery. Once the pilus extends into the extracellular space, with VirB5 at its extremity, the T4SS scans the environment for a recipient cell. Upon contact, the tip of the pilus must either punch through or/ and assemble a pore to form a channel in the recipient membrane (Low et al, 2022). The exact mechanisms of this stage remain unclear. However, it is thought that the contact with the recipient triggers a signal that activates the secretion mode of the T4SS, leading to reorganisation of the machinery for substrate recruitment and secretion (Hu et al, 2019). While structural data for this step is lacking, a plausible model suggests that VirD4, initially sitting outside the IMC, moves centrally to hexamerise in place of VirB4. This ATPase shift could allow VirD4 to accomplish substrate recruitment and translocation functions (Ilangovan et al, 2015). The substrate then passes through the T4SS, the pilus and ultimately reaches the recipient cell's cytoplasm, where further processes conduct to the establishment and activities of the secreted substrate.

## Assembly of the T4SS machinery

The construction of this complex nanomachine is realised through a precise sequence of events, where individual components are assembled into subcomplexes and ultimately form a complete and active system (Hu et al, 2019; Ghosal et al, 2019; Khara et al, 2021). The process begins with the export of VirB1 to the periplasm,

where it hydrolyses the peptidoglycan layer to create a localised space for T4SS assembly (Ward et al, 2002). VirB1 also interacts with the first components to be assembled, i.e, VirB8, VirB9, and VirB10 (Höppner et al, 2005; Ward et al, 2002). Thus, VirB1 plays a dual role in peptidoglycan degradation and the proper localisation of T4SS assembly. Although the regulation of VirB1 activity is not fully understood, its peptidoglycan lysis function must cease at a certain point to prevent excessive degradation, which could compromise the structural integrity of the bacterial cell wall (Bayer et al, 2001).

Once the space is prepared, the assembly of the OMCC begins, consisting of VirB7, VirB9 and VirB10. As the next step, the Stalk, composed of VirB5 and VirB6, is centrally positioned and likely assembles before the Arches, which are formed by VirB8 and encircle the Stalk (Ghosal et al, 2019). VirB10, through its Arches and IM domains, coordinates the assembly by interacting with the Arches and the Stalk, respectively (Macé and Waksman, 2024). The final subcomplex to be assembled is the IMC, made of VirB3 and VirB4. In particular, VirB4 forms a dimer, and multiple dimers assemble to form the iconic hexamer structure of the IMC, which is connected to the rest of the T4SS by interacting with VirB8$_{tails}$ and VirB10$_{cyto}$ domains (Macé and Waksman, 2024). Once the IMC is in place, the assembly of the T4SS is finalised when VirB11 binds to the bottom of the IMC, stabilising the double hexameric ATPase complex formed by VirB4 and VirB11 (Khara et al, 2021; Park et al, 2020). Following the assembly, the T4SS undergoes specific post-assembly modifications, including the formation of both disulfide bonds by enzymes encoded within the T4SS itself (Jameson-Lee et al, 2011; Apostol et al, 2024), and establishment of the symmetry mismatch between the O-layer and I-layer of the OMCC, although the mechanism leading to this remains unknown (Costa et al, 2024).

## Mechanism of pilus biogenesis

Upon assembly and activation, the T4SS initiates pilus production by forming a helical structure with fivefold symmetry composed of numerous VirB2-lipid subunits (Costa et al, 2016), with VirB5 at its extremity (Aly and Baron, 2007). VirB2 pilins are initially embedded in the inner membrane, near VirB3, awaiting extraction along with a specific lipid for subsequent assembly into the pilus. Co-evolution studies have identified VirB3 and VirB6 as binding partners for VirB2, facilitating its recruitment (Macé et al, 2022b) (Fig. 2A). These predicted interactions occur within the transmembrane domains (TMs) of VirB3 and VirB6, consistent with VirB2's location in the IM. The proximity of VirB3 and VirB6 TMs allows them to simultaneously interact with VirB2, designating this region

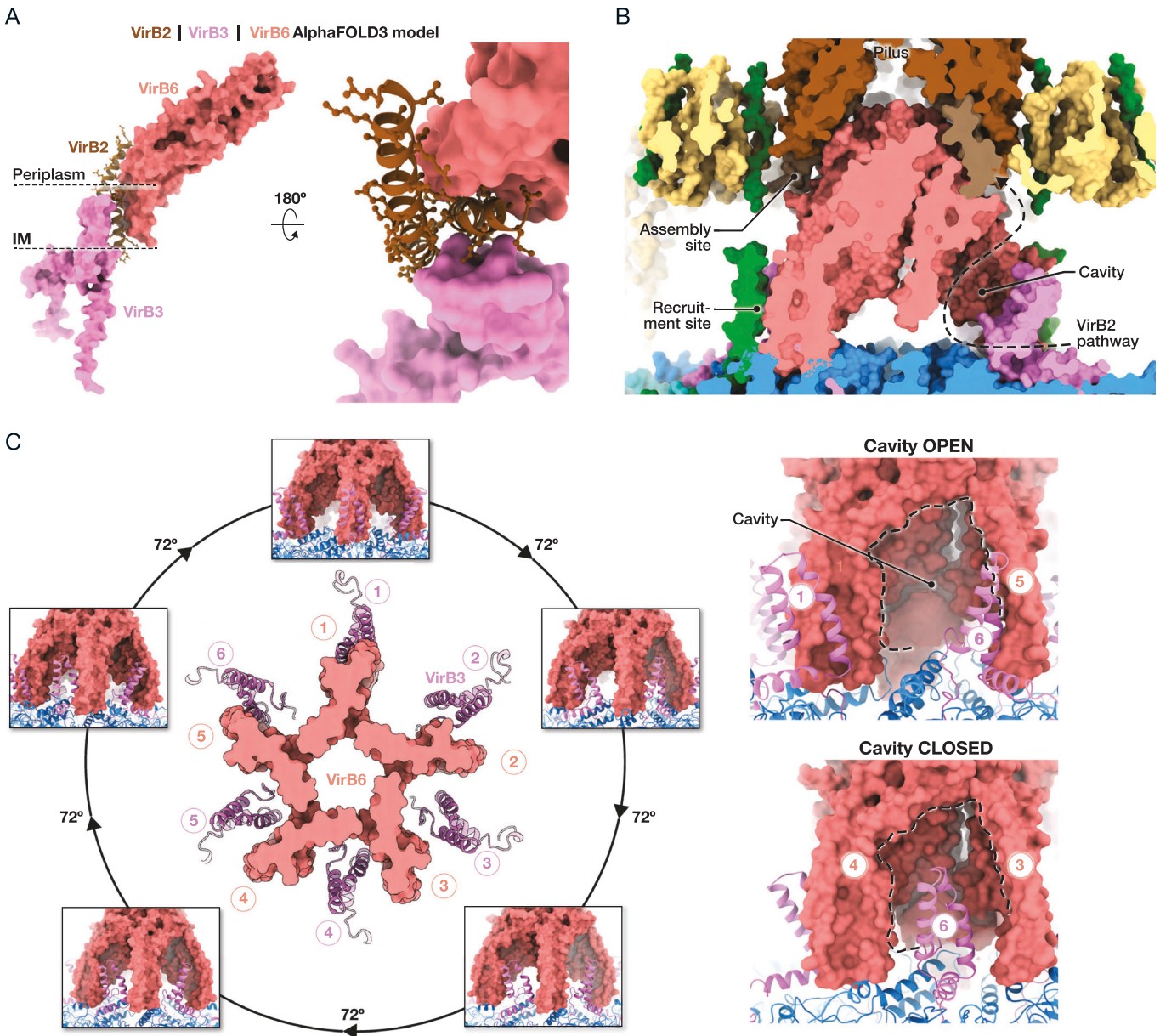

**Figure 2.** VirB2 pilin to pilus and IMC/STALK mismatch symmetry.

(A) Alphafold3 model of VirB2 recruitment by VirB3-B6. Two views of VirB2 predicted to interact with both VirB3 and VirB6. VirB2 is displayed on ribbon with side-chains in stick representation, while VirB3 and VirB6 are shown in surface. This interaction and recruitment model are based on a previous co-evolution study (Macé et al, 2022b). (B) VirB2 pathway to the pilus through the recruitment cavity. Cut-view of the T4SS assembly with the pilus integrated according to Fig. 1. The model structure is represented in surface and colour-coded by protein. The dashed line indicates the proposed pathway of VirB2, through the cavity that connects the recruitment and assembly sites. (C) Overview of mismatch symmetry organisation. Left, top view of the VirB6 pentameric structure from the STALK (PDB ID 8RT9) shown in surface and red, and the VirB3 hexameric structure from the IMC (PDB ID 7O41) shown in ribbon and pink. This top view is encircled by five front-views (rotated 72° each) illustrating the VirB3-VirB6 mismatch organisation. Right, zoom-in on two front-views showing the cavity (highlighted by dashed lines) in both open or closed states.

as the "VirB2 recruitment site" (Macé et al, 2022b) (Fig. 2B). However, in the resting-state cryo-EM structure, this site is obstructed by VirB10$_{IM}$, which likely blocks pilus biogenesis until it moves away to enable VirB2 binding and initiation of the pilus biogenesis process (Macé and Waksman, 2024).

The VirB2 recruitment site is strategically positioned at the Stalk/IMC mismatch symmetry (pentameric versus hexameric, respectively) (Fig. 2C, left). This site features six VirB3 units and

five VirB6 units. The exact arrangement and the role of this mismatch remain unclear, as the high-resolution cryo-EM structure for this region has not been fully solved. Nevertheless, this mismatch is thought to play a key role in VirB2 recruitment and support the highly dynamic assembly process of pilus biogenesis. Structurally, VirB6 and VirB3 form a cavity (Fig. 2C, right), where VirB3 acts as an entry-gate from the inner membrane, while the cavity's exits point towards the pilus assembly site in the periplasm,

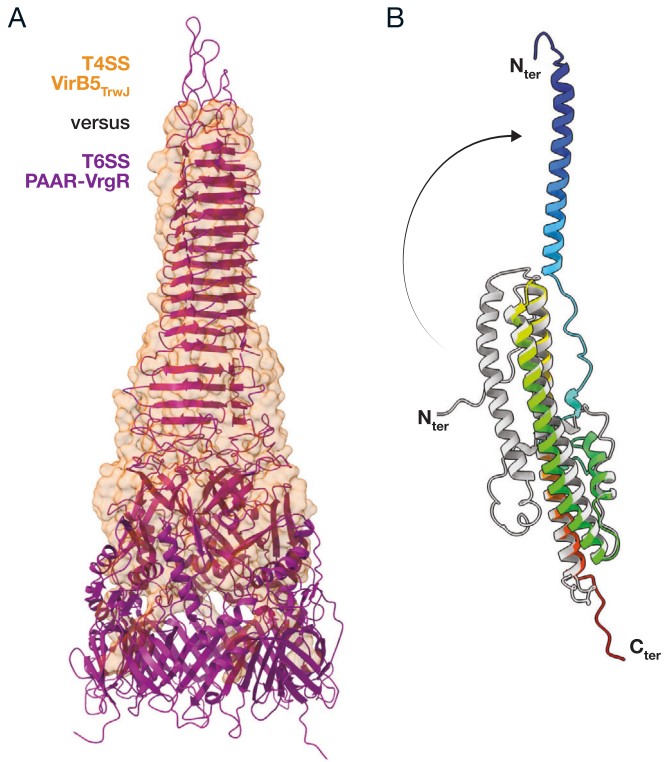

**Figure 3. Pilus-tip comparison and VirB5 conformation change.**

(A) Structure superposition of T4SS and T6SS tips. The T4SS tip (VirB5_TrwJ— PDB ID 8RT9) is shown in an orange transparent surface, while the T6SS tip (PAAR/VgrG—PDB ID 4JIW/PDB ID 8GRA) is shown in a purple ribbon representation. This panel illustrates the shape similarity between both tips, suggesting a probable analogous function in piercing the recipient membrane. (B) VirB5 conformation change. Two structures of VirB5 are presented in ribbon. The monomeric conformation is shown in grey (VirB5_TraC PDB ID 1R8I), and a protomer of the pentameric conformation is displayed in a rainbow gradient (VirB5_TrwJ—PDB ID 8RT9). The arrow indicates the movement of the N-terminal helix of VirB5 between the two conformations.

positioned between the Stalk and the Arches (Costa et al, 2024). The location of both the VirB2 recruitment site and pilus assembly site may indicate that VirB2 travels through this cavity (Macé et al, 2022b). The force required to extract VirB2-lipid subunits from the inner membrane is provided by the VirB4-VirB11 ATPases (Kerr and Christie, 2010), although the precise mechanism of energy conversion into pilin extraction remains unknown.

The exact details of pilus assembly are still being elucidated, but a recent model proposes that formation occurs at the pilus assembly site located between VirB5 and VirB6 (Macé et al, 2022b). At this site, VirB2-pilin subunits are assembled into a stable, hydrophilic polymer that forms the pilus (Costa et al, 2016). The assembly site is tightly surrounded by the Arches, composed of VirB10 and VirB8, suggesting that spatial constraints within this region may facilitate the transition of VirB2-lipid subunits into the pilus assembly. The pilus then grows beneath the VirB5 tip complex, which is pre-assembled on top of VirB6, and pushes upwards to open the initially closed OMCC gate formed by a VirB10 homo-oligomer (Chandran et al, 2009). Thus, VirB10 plays a central role in regulating pilus biogenesis. Its functions range

from sensing ATPase activity through its interactions with VirB4 in the cytoplasm and coordinating VirB2 recruitment in the inner membrane, to managing pilus assembly in the periplasm, and controlling the passage of the pilus through the outer membrane (Macé and Waksman, 2024) (Fig. 1D).

Pilus biogenesis is essential for T4SS functionality, as the pilus mediates the critical interaction with the recipient cells. Inactive pilus production leads to a dysfunctional T4SS (Costa et al, 2016), underscoring the necessity of pilus formation for the secretion process. While the role of the pilus in contacting recipient cells is established, the exact mechanism by which it interacts with or penetrates the recipient membrane remains poorly understood.

## Donor–recipient cell interaction

Given its position at the pilus tip, VirB5 is thought to play a central role in the interaction of T4SS with recipient cells. Structurally, VirB5 assembles into a robust pentamer with the largest protein–protein interface within the T4SS, stabilised by ions and yet unidentified ligands. While the helical structure of VirB5 differs from the β-stranded PAAR-VgrG protein that forms the tip of the Type VI secretion systems (T6SS) (Shneider et al, 2013), it exhibits a similar overall shape and structural stability properties (Fig. 3A). In addition, VirB5 has structural similarity with pore-forming and sensor proteins (Macé et al, 2022b), suggesting potential analogous activity. Proteins in these families typically undergo conformational changes upon contact, particularly in their N-terminal helical regions, consistent with structural differences observed in VirB5's monomeric and pentameric forms (Yeo et al, 2003) (Fig. 3B).

To establish a continuous secretion pathway, the T4SS machinery must overcome the recipient cell's outer membrane and possibly penetrate deeper into the peptidoglycan layer and inner membrane. From the structural data, two primary hypotheses regarding the role of VirB5 have emerged: either VirB5 forms a membrane pore or it acts as a cell wall-piercing device (Backert et al, 2008). Despite extensive studies, no specific T4SS receptor has been identified (Pérez-Mendoza and de la Cruz, 2009), suggesting that VirB5 may not require specific binding partners or that it may encounter multiple receptors across different host cells, complicating their identification. Given these considerations, VirB5 may act similarly to the T6SS Paar protein, serving as a "drill bit" for the T4SS machinery. It may also undergo conformational changes upon interaction with membrane or periplasmic environments, enabling it to form a pore or function as a "grappling hook". The stability of donor–recipient contact during secretion, which can last up to an hour (Goldlust et al, 2023), is later reinforced by adhesin-receptor interactions that provide more robust cell-to-cell adherence (Low et al, 2022).

### T4SS adhesion

The pilus tip, composed primarily of VirB5, acts as the main adhesin in T4SS-mediated contact. In fact, a study has shown that bacterial conjugation can occur across distances where only the pilus connects the two cells (Goldlust et al, 2023; Babić et al, 2008). Additional surface adhesins contribute to Mating Pair Stabilisation (MPS) after pilus retraction (Costa et al, 2024). These accessory adhesins, which vary across systems, increase conjugation efficiency

by recognising host molecules, such as sugars or proteins on the recipient cell surface (Low et al, 2022). Though not essential for DNA transfer, they significantly increase conjugation efficiency (Low et al, 2022). For instance, proteins from the KikA-Pep family are exported to the donor cell surface, where they interact closely with VirB5 to form higher-order complexes that enhance conjugative transfer (González-Rivera et al, 2019). In the F-type T4SS, TraN stabilises the mating pair by interacting with outer membrane proteins (OMPs) on the recipient cell surface and provides species specificity (Low et al, 2022). In more evolved systems, such as *Helicobacter pylori* T4SS$_{Cag}$ system, additional proteins, such as CagI and CagY, are associated with the surface-exposed structures and contribute to specific interactions with human gastric cells (Kwok et al, 2007; Pham et al, 2012; Tegtmeyer et al, 2020). However, their presence at the pilus tip remains to be clearly demonstrated.

## Exclusion mechanism

Conjugative plasmids encode exclusion mechanisms to prevent redundant conjugation events between cells carrying related plasmids. These exclusion mechanisms are subclassified as surface exclusion or entry exclusion, depending on whether they interfere with MPS or the DNA transfer process (Garcillán-Barcia and de la Cruz, 2008). Surface exclusion proteins, such as those in the TraT prototype family, are highly conserved and prevent initial contact with the recipient cell. TraT in F-plasmid forms a "basketball hoop" structure anchored in the outer membrane and facing the extracellular environment, suggesting that it could block pilus entry by physically trapping it (Seddon et al, 2024). Recently, other surface exclusion proteins in the Sfx family were proposed to target TraN specifically, blocking its interaction with outer membrane proteins and thereby inhibiting conjugation (Rivard et al, 2024). Because of these properties, surface exclusion proteins are attractive targets for the development of conjugation inhibitors (Getino and De La Cruz, 2018).

Conversely, entry exclusion proteins, like those in the TraS prototype family, interfere with later stages of conjugation, such as pore formation or DNA secretion. TraS encoded by the F-plasmid localises in the inner membrane of the recipient cell and interacts with TraG$_{B6}$ from the donor cell (Audette et al, 2007). The TraG$_{B6}$-TraS interaction may interfere with the conjugative signalling events that normally trigger DNA transfer, although further analysis is required to confirm this hypothesis (Costa et al, 2024).

In conclusion, the donor–recipient cell interaction likely acts as a transition signal to shift the T4SS into secretion mode. However, how a distant interaction at the pilus tip translates into a signal sensed at the T4SS core remains unclear. Understanding the long-distance nature of this interaction and its role in initiating secretion at the base of the machinery remains a central question in the research field.

# The substrate recruitment and translocation mechanisms

The mechanisms underlying substrate recruitment by T4SS remain elusive, largely due to the difficulty of capturing these rapid and transient events, as well as the diversity of recruitment processes across different T4S systems. The following section summarises the available insights into these processes.

## Inner membrane complex reorganisation

Upon contact with the recipient cell and receiving a corresponding signal, the T4SS machinery undergoes structural reorganisation to shift into secretion mode. Particularly within the inner membrane complex (IMC), where cryo-EM structural studies suggest that this reorganisation is a dynamic process involving VirB4 (Macé et al, 2022b; Low et al, 2014) (Fig. 4). In the T4SS resting-state structure, VirB4 adopts a "hexamer of dimers" configuration; however, in its free-complex form, VirB4 arranges into a "trimer of dimers" (Macé et al, 2022b). These two distinct quaternary structures of VirB4 suggest that the IMC assembly can dynamically disassemble and reconfigure as required. The 12 VirB4 subunits can thus alternate between two configurations: either as a central hexamer of dimers, or as two peripheral hexameric assemblies (Fig. 4).

The exact mechanism of T4SS reorganisation remains unclear, although VirB10 is a strong candidate for initiating the switch, given its interactions with both ATPases, VirB4 and VirD4, as well as the pilus (Macé and Waksman, 2024; Llosa et al, 2003). Precisely, structural studies have identified 12 VirB10$_{Arches}$ subunits, six VirB10 (referred to as VirB10$_{Arches-A}$) interacting with VirB4 at the IMC, while the remaining six (VirB10$_{Arches-B}$) arrive at the IM and could potentially interact with VirD4 (Macé and Waksman, 2024). Altogether, the activation signal likely induces conformational changes in the N-terminal domains of VirB10 to promote IMC reorganisation. These structural observations align with previous studies suggesting VirB10's role as an ATPase sensor (Cascales and Christie, 2004). In this model, the VirB4 dimer functions like a "Lego piece", allowing flexible IMC reorganisation. Simultaneously, VirD4, initially positioned peripherally in a monomeric form (Redzej et al, 2017) assembles into a centralised hexameric conformation (Gomis-Rüth et al, 2001), ready to initiate substrate recruitment. In essence, the signal-induced switching mechanism through the VirB10 N-terminus may enable the centralised ATPase to transition from one role (VirB4's involvement in pilus biogenesis) to another—the role of VirD4 in substrate recruitment and translocation. This reorganisation model is based on structural studies and remains to be confirmed experimentally.

## Molecular recognition of the substrate

Following IMC reorganisation, substrate recruitment by the T4SS machinery begins. Although T4SS types are often classified based on their secreted substrates, either DNA in conjugative transfer or protein effectors in host-pathogen interactions, the primary recruited substrate is consistently a protein (Meir et al, 2023). Indeed, in conjugative systems, DNA is indirectly recruited through linkage to a relaxase protein, which, along with additional relaxosome components, are the direct targets of the type 4 coupling complex (T4CP) (Redzej et al, 2013a; Lang et al, 2010; Lu et al, 2008). The coupling complex may consist solely of VirD4 protein (Wu et al, 2023) or include additional components forming an extended recruiting platform (Kwak et al, 2017). VirD4 is an AAA+ ATPase and functionally active as a hexameric complex (Fig. 1C), although it also presents as a monomeric form (Gomis-Rüth et al, 2001). Two VirD4 domains are responsible for substrate

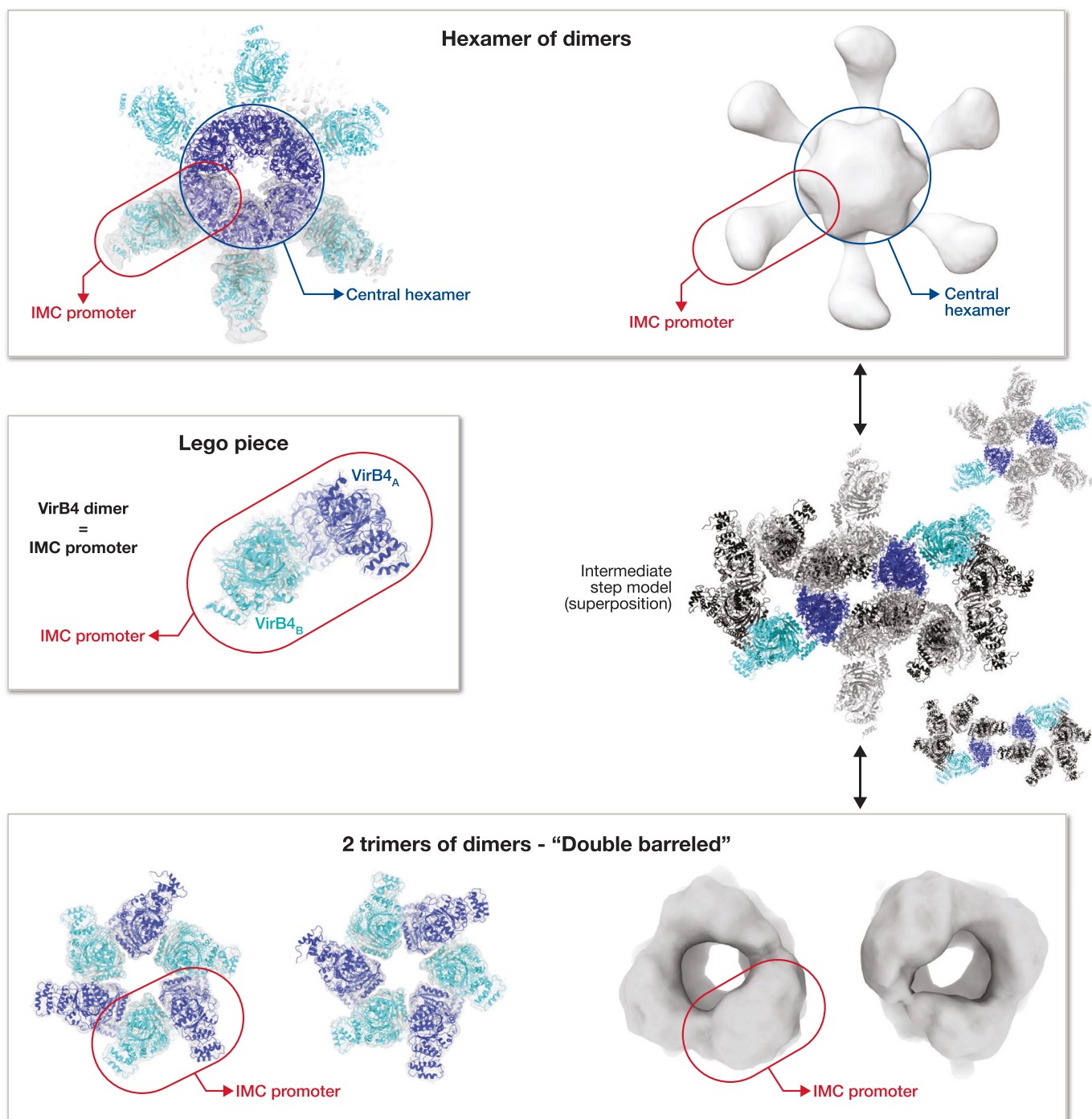

**Figure 4. VirB4 dimer as a Lego piece.**

Various conformational structures illustrate the VirB4 dimer's function as a Lego piece. From the VirB4-dimer, the unit Lego piece (shown in centre-left) allows for the global architecture of the IMC to reorganise into two conformations, (i) hexamer of dimers (observed via cryo-ET (Park et al, 2020) & cryo-EM (Macé et al, 2022b)) and (ii) two trimers of dimers (observed by NSEM (Low et al, 2014; Peña et al, 2012) and cryo-EM (Macé et al, 2022b)). The transition is illustrated by a proposed intermediate structural step, shown in the centre-right, based on the superposition of both conformations onto two common VirB4 dimers, or "Lego pieces," which are colour-coded for clarity. Relevant structural data: (i) Lego piece: PDB ID 7OIU—EMDB ID 12933; (ii) hexamer of dimers: PDB ID 7O41—EMDB ID 13767; (iii) two trimers of dimers: PDB ID 7O42—EMDB ID 2567.

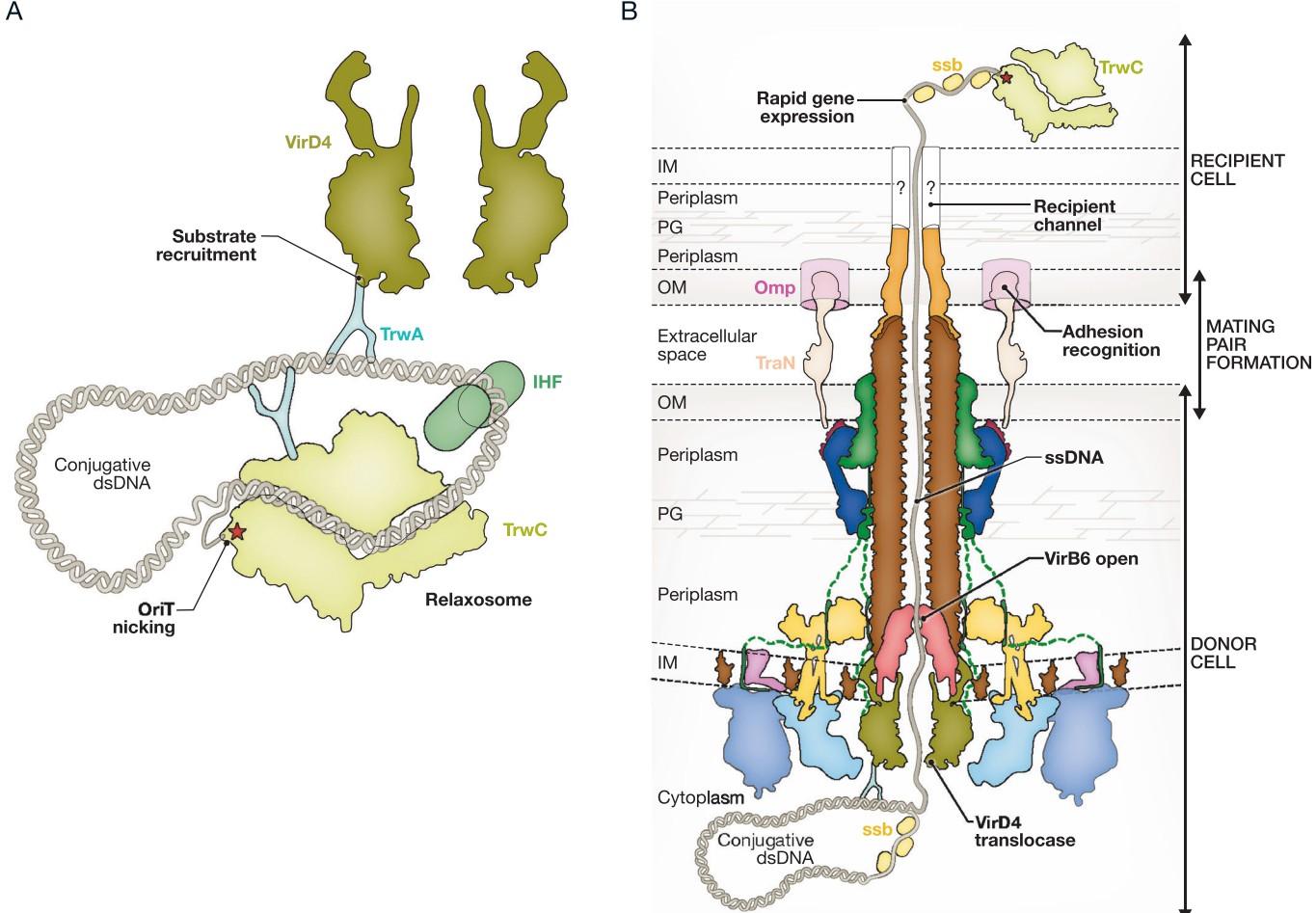

**Figure 5. The substrate pathway.**

(A) Schematic representation of T4SS substrate recruitment. The relaxosome, composed of double-stranded DNA (dsDNA), along with the relaxases TrwC, TrwA and IHF, is recruited by the T4SS *via* VirD4, which interacts with TrwA. (B) Schematic representation of T4SS during secretion, with the VirD4 ATP-dependent translocase pushing single-stranded DNA (ssDNA) through VirB6, the pilus and VirB5 into the cytoplasm of the recipient cell. During this process, adhesin interactions, such as TraN/Omp, facilitate the formation of a mating pair between donor and recipient cells. The ssDNA is protected by the ssb protein in both bacteria, and the T4SS leading genes that first arrive in the recipient cell are rapidly expressed (Samuel et al, 2024).

recruitment: the all-alpha domain (AAD) positioned at the base of the hexamer, and the C-terminal tail of VirD4 when such is present (Oka et al, 2022). The VirD4 tail domain interacts with the substrate either directly or *via* additional components linked to the VirD4 tail (Lu et al, 2008; Kwak et al, 2017). The exact positioning of VirD4 within the T4SS remains unclear, as it has only been observed at low resolution (Redzej et al, 2017), with its single confirmed interaction being with the N-terminal domain of VirB10 at the inner membrane (Llosa et al, 2003). Further molecular insights are needed to clarify the role of this interaction and to definitively establish VirD4's position relative to the rest of the T4SS machinery.

### Relaxosome recruitment

In conjugative T4SSs, the relaxosome complex is the target for recruitment (Fig. 5A). In the prototype F-plasmid system, the relaxosome consists of (i) DNA containing a 200–300 basepair OriT region (Ares-Arroyo et al, 2024), (ii) TraI, a large protein

known as relaxase (~1000 amino acids) (Guasch et al, 2003), and three accessory proteins: (iii) TraY, a small DNA-binding protein (Nelson and Matson, 1996), (iv) IHF, a homing DNA-bending protein that induces a 180° turn in the DNA (Rice et al, 1996), and (v) TraM, a tetrameric DNA-binding protein (Wong et al, 2011).

Relaxase protein exhibits significant diversity across T4SSs (Ilangovan et al, 2015, 2017), due to which they are not included in Table 1. Relaxosome assembly involves specific interactions between these proteins and DNA motifs within the OriT region. While the components of the relaxosome are well-characterised, the precise conditions required for their assembly remain unclear. It is still uncertain whether the relaxosome assembles in proximity to the T4SS in anticipation of a secretion signal or only after the signal is received. Once assembled, the relaxosome induces conformational changes in the DNA topology that expose the nick (nic) site. The relaxase cleaves this site and covalently attaches to the resulting single-stranded DNA (ssDNA, referred to as T-DNA for "transferred-DNA") *via* a tyrosine residue (Ilangovan et al, 2015, 2017).

The recruitment mechanism of the relaxosome by the coupling complex remains only partially understood. Two relaxase domains, TSA and TSB (for translocation signal A and B), are involved in recruitment through interactions with VirD4 (Lang et al, 2010; Redzej et al, 2013b). However, the lack of structural and additional mechanistic studies prevents the formulation of a definitive recruitment model. In addition, a weak interaction surface has been observed between TraM and VirD4 (Lu et al, 2008). This interaction likely represents one step in a more complex recruitment process, as TraM itself is not the final substrate; instead, relaxase-ssDNA complex is the ultimate substrate delivered by the T4SS.

### Protein effector recruitment

The T4SS$_{Dot/Icm}$ of *Legionella pneumophila* represents the most complex T4CP identified to date, comprising seven proteins, including IcmS and IcmW as integral subunits of the T4CP, and the chaperone LvgA, which transiently associates with the complex to mediate selective effector recruitment (Macé et al, 2022a; Meir et al, 2023, 2020). This extensive platform supports a remarkably diverse substrate repertoire, enabling recruitment of over 300 distinct effector proteins through at least three unique recognition processes (Burstein et al, 2016). Despite its large structure, interaction between this T4CP and the broader T4SS machinery was never observed.

In contrast, *Xanthomonas citri* employs a minimised T4SS for bacterial competition, composed only of VirD4 within its T4CP. Secreted proteins possess a conserved N-terminal domain termed XVIPCD, structurally characterised and proposed to be recognised by the all-alpha domain (AAD) of VirD4 (Oka et al, 2022). Another example of T4CP made of one protein, the VirD4$_{Cagβ}$ of the T4SS$_{Cag}$ system in *Helicobacter pylori*, recruits and secretes a single protein, CagA, which induces oncogenic transformation in human gastric cells (Saadat et al, 2007; Hatakeyama, 2004). The all-alpha domain (AAD) of VirD4$_{Cagβ}$ interacts with the CagA translocation factor CagZ before hexamerisation (Wu et al, 2023). In addition, CagA requires a chaperone, CagF, for efficient recruitment (Couturier et al, 2006). Although the structures of these components have been solved, the exact molecular mechanism underlying CagA recruitment and secretion remain undefined. Finally, certain conjugative plasmids can secrete accessory proteins beyond the relaxase–ssDNA complex (Al Mamun et al, 2024), highlighting the T4SS versatility in handling various substrates.

In T4SS, as in all secretion systems, precise substrate recognition is essential to ensure that only the intended substrates are recruited and translocated, thereby maintaining system functionality. Achieving this specificity in recruitment likely involves multiple steps to enhance fidelity and accuracy in substrate selection.

### The translocation mechanism

Substrate translocation is primarily mediated by the ATP-dependent VirD4 translocase protein. This process begins with the substrate entering into the central channel of VirD4 from the cytoplasm. ATP hydrolysis provides the energy needed to drive conformational changes in VirD4 subunits, which induce a sequential shift in lysine residues within the channel (de Paz et al, 2010). These movements result in a stepwise transport of the substrate, whether protein or DNA. Our understanding of these

molecular mechanisms primarily stems from studies on FtsK family homologues, a family of double-stranded DNA translocases to which VirD4 belongs (Iyer et al, 2004; Rzechorzek et al, 2014; Jean et al, 2020). Based on AlphaFold models and partial cryo-EM structures, VirD4 is predicted to include an additional transmembrane domain that may extend its central channel to form a continuous translocation pathway across the inner membrane (Gomis-Rüth et al, 2001, Wu et al, 2023). Beyond this exit point, the specific route and interactions guiding the substrate from VirD4 into the periplasm remain undefined (Costa et al, 2024). In addition, in all T4SS structural studies, the T4CP has never been observed under active hexameric form, leaving the exact mechanism by which VirD4 recognises and translocates both DNA and protein substrates still unclear. Furthermore, emerging evidence suggests that VirB4 may have direct DNA-binding capabilities (Peña et al, 2012; Li et al, 2012). This raises the intriguing possibility that VirB4 could contribute to substrate stabilisation or assist in the transfer of DNA to other components of the system. These hypotheses highlight the need for further investigation to elucidate the precise molecular mechanisms underpinning substrate recruitment and transfer within the T4SS.

### The substrate pathway

The complete organisation of the T4SS in its secretion mode remains largely unknown, leaving the substrate's pathway through the machinery incompletely characterised. Nonetheless, available structural data allow us to propose a plausible post-VirD4 translocation pathway model. After exiting VirD4, the substrate likely travels through the T4SS apparatus toward the pilus, as recent studies confirm that substrates traverse the pilus during secretion (Goldlust et al, 2023). Centrally positioned within the T4SS, the base of the pilus is modelled to sit atop VirB6 (Macé et al, 2022b; Costa et al, 2024). Structurally, VirB6 forms a funnel-shaped exit oriented toward the pilus lumen, suggesting that the substrate has to pass through VirB6 to enter the pilus (Figs. 1d and 5B).

Additionally, indirect pathways may also exist, particularly in more complex T4S systems. For example, T4BSS$_{Dot/Icm}$ in *Legionella pneumophila* uses an alternative pathway involving specific effectors containing strongly hydrophobic transmembrane domains to cross the inner membrane (Malmsheimer et al, 2024). In summary, like the "heart" of T4SS, VirD4 serves as the pump driving secretion, while VirB6 functions as a valve, regulating substrate entry into the pilus, together facilitating substrate progression through the T4SS machinery.

### Conclusion of the substrate recruitment and translocation mechanisms

The T4SS substrate journey starts with recruitment and translocation mediated by VirD4, continues through VirB6, then enters into the pilus, and ultimately reaches the recipient cell by crossing its membrane, likely *via* VirB5 or other components forming a translocation pore (Fig. 5B). Throughout this process, additional host proteins, such as the ssb protein, contribute to the stability and functionality of the substrate within both donor and recipient bacteria (Couturier et al, 2023). Upon arrival in the recipient cytoplasm, transferred DNA initiates rapid gene expression, often activating crucial anti-defence mechanisms (Samuel et al, 2024).

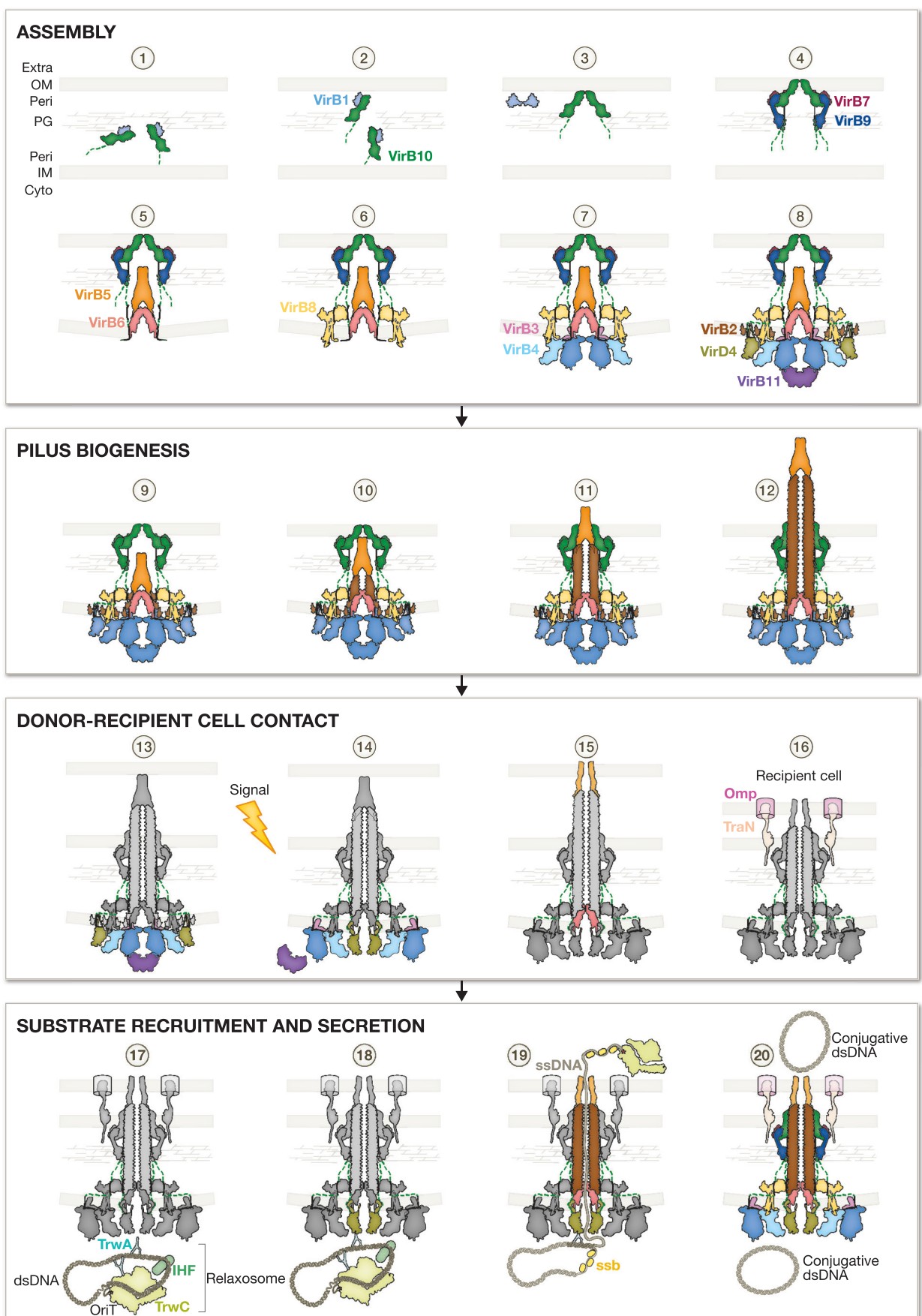

**Figure 6.  Step-by-step representation of the T4SS lifecycle.**

The T4SS functional cycle is presented in chronological order and colour-coded by proteins as in Fig. 1, with each key process segmented: assembly (steps 1–8), pilus biogenesis (steps 9–12), donor–recipient cell contact (steps 13–16), and substrate recruitment and secretion (steps 17–20). All steps depicted are based on literature-derived models and structural predictions, and have not been directly observed experimentally, except for step 7, which has been resolved by cryo-EM (Macé et al, 2022b; Macé and Waksman, 2024), and step 12, which has been visualised by cryo-electron tomography (Hu et al, 2019). ASSEMBLY: Step 1, the VirB1-VirB10 complex reaches the peptidoglycan (PG), where VirB1's lytic activity begins. Step 2, VirB10 accesses the outer membrane (OM) through VirB1's opening hole. Step 3, VirB10 monomer assembles, while VirB1 forms an inactive homodimer. Step 4, VirB7 and VirB9 integrate, completing the OMCC subcomplex assembly. Step 5, the Stalk (VirB5 and VirB6) is inserted centrally *via* VirB6-VirB10 interaction in the inner membrane (IM). Step 6, the Arches are formed by VirB8$_{peri}$, with VirB10 anchoring the assembly. Step 7, the IMC is assembled in the cytoplasm with the VirB4 dimer ("Lego piece") connecting to the larger T4SS structure *via* VirB8 and VirB10 cytoplasmic domains. Step 8, T4SS assembly is completed as both cytoplasmic ATPases, VirB11 and VirD4, join at the IMC's base and periphery, respectively. PILUS BIOGENESIS: Step 9, VirB2 pilins in the IM are recruited by VirB3 and VirB6. Step 10, ATPase-dependent energy from VirB4 and VirB11 extracts VirB2 pilins to assemble the pilus. Step 11, iterative VirB2 addition grows the pilus, with VirB5 at the tip, allowing it to extend through the OMCC. Step 12, the pilus is now extracellular, ready for recipient cell contact. DONOR–RECIPIENT CELL CONTACT: Step 13, pilus contact with the recipient cell membrane triggers a signal. Step 14, this signal induces an IMC reorganisation, centralising and activating VirD4 in hexameric form, while VirB4 and VirB11 ATPases are inactivated. Step 15, the reorganisation completes with VirB5 and VirB6 opening, clearing the substrate pathway. Step 16, interactions such as with entry exclusion proteins may inhibit secretion, while adhesins such as TraN/Omp stabilise secretion by forming a mating pair. SUBSTRATE RECRUITMENT & SECRETION: Step 17, the relaxosome assembles at the OriT site, where TrwC relaxase nicks the dsDNA, being covalently linked to the single transferred DNA strand (ssDNA). Step 18, VirD4 interaction recruits the relaxosome *via* TrwA. Step 19, VirD4's ATP-dependent activity pushes the relaxase and the ssDNA through VirB6, the pilus, and VirB5, delivering it to the recipient cell cytoplasm. Step 20, once arrived, leading genes on the ssDNA are expressed, facilitating defence, stabilisation, and plasmid replication—readying the plasmid for further dissemination within the bacterial population.

While significant progress has been made, further research is essential to fully clarify the molecular specificities of substrate recruitment and translocation across T4SS and recipient cell membranes.

## Concluding remarks and future perspectives

Our exploration of the T4SS through its key functional stages—machinery assembly, pilus biogenesis, donor–recipient interaction, and substrate recruitment and secretion— highlights the complexity inherent in studying this multifaceted bacterial apparatus. As illustrated in Fig. 6, T4SS operates through a series of interconnected sub-mechanisms, each with potential intermediate states that add layers of complexity to our understanding of how these systems function. Central to the switching between these sub-mechanisms, VirB10 emerges as a key regulatory protein. The pivotal role of VirB10 in coordinating the functional transitions is essential for T4SS operation.

Despite recent advancements, several aspects of T4SS remain to be fully deciphered. Key challenges include elucidation of pilus-recipient cell interactions, identification of the precise triggering signals, and clarification of the organisation of the T4SS machinery during substrate recruitment and translocation. These gaps underscore the need for further investigation to achieve a comprehensive understanding of T4SS functionality. Advanced structural approaches, such as cryo-correlative light and electron microscopy (CLEM) (Nogales and Mahamid, 2024), combined with cutting-edge fluorescence techniques to spot conjugation events (Goldlust et al, 2023; Couturier et al, 2023), hold great promise for addressing these critical questions. Last but not least, these insights offer a promising foundation for translational applications. Engineered T4SS holds potential for therapeutic delivery, while T4SS-targeting strategies could pave the way for novel antimicrobial agents against drug-resistant pathogens.

## Peer review information

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

## Acknowledgements

The authors express their gratitude to Prof. Gabriel Waksman and Prof. Reynald Gillet for their substantial input and invaluable advice. We also thank many researchers in the field of Type IV Secretion Systems, whose extensive work has enriched the literature, much of which could not be exhaustively covered in this overview. This work was supported by France 2030 under the Agence Nationale de la Recherche (ANR-22-PAMR-0005) and the ANR Tremplin-ERC VIRULENSSE.

## Author contributions

**Pierre Paillard**: Data curation; Validation; Investigation; Visualisation; Writing —review and editing. **Quentin Rouger**: Formal analysis; Validation; Investigation; Writing—review and editing. **Manon Thomet**: Writing—review and editing. **Kévin Macé**: Conceptualisation; Supervision; Funding acquisition; Visualisation; Writing—original draft; Project administration; Writing—review and editing.

## Disclosure and competing interests statement

The authors declare no competing interests.

