## [Peer Review File · The EMBO Journal]

Type IV Secretion Systems: from structures to mechanisms

Kévin Macé, Pierre Paillard, Quentin Rouger, and Manon Thomet

Corresponding author: Kévin Macé (kevin.mace@univ-rennes.fr)

Review Timeline:

Submission Date:	11th Mar 25
Editorial Decision:	14th May 25
Revision Received:	28th May 25
Editorial Decision:	4th Jul 25
Revision Received:	15th Jul 25
Accepted:	16th Sep 25

Editor: Ieva Gailite

Transaction Report:

Dear Kevin,

Thank you for submitting your review article to The EMBO Journal, and I am sorry for the protracted manuscript evaluation process due to delayed report submission and conference travel. Your manuscript has now been seen by two reviewers, and I have attached their comments below.

As you will see, both reviewers appreciate the topic and the timeliness of the review. In addition, they provide several constructive points for the improvement of the article and the figures, which reviewer #1 has also included in the manuscript file that can be accessed here due to its size: <https://oc.embl.de/index.php/s/SCKnwMwE3YnmXR7>. Therefore, I would invite you to submit a revised version of the manuscript with these suggestions incorporated.

From the editorial side, I have also included below further details on figure preparation for the final version. Finally, we do not allow Supplementary Figures for reviews, and I would therefore recommend adding the corresponding database references into the Table 1. I am happy to discuss the possible ways of incorporating this information further.

Thank you for preparing such a timely review, and I look forward to receiving the final version.

With best wishes,

Ieva

Instructions for figure preparation:

We work with Somersault1824 for graphics support for figures. They will assist with getting the figure to a publication ready state. What we need from you is a draft that accurately illustrates the key scientific concepts that you wish to show. Please also ensure that the figure draft is conceptually as close as possible to the final version as we cannot offer to do multiple rounds of revisions (and substantial changes might necessitate a complete redesign).

Please also note the following points:

- If there are certain aspects of your figure draft that are based upon assumptions or where the scientific data remains ambiguous (for example, schematically depicting a presumed direct protein-protein interaction, protein shape or subcellular localizations etc.) please add a comment so that we can work with you on an accurate depiction. Please ensure the directionality and nature of interactions is presented accurately.
- If the figure or single panels of the figure have been adapted from a published figure, please add this information to the figure legend (e.g., 'Adapted from...' or 'Based on...'). The editor will discuss if a reference and permission will be necessary.
- Please only re-use figures or parts of a figure if this is essential for understanding the concept communicated. Often a reference to a previous paper will suffice. If the figure contains re-used images or elements of images, including schematics, micrographs or photos, please make sure that you have the permission/license to publish it (this also applies to your own previous work, if the journal you published in retains copyright. Certain 'creative commons' open access licenses, such as CC-BY 4.0, allow re-use without additional formal permissions). All re-used material must be explicitly cited.
- If you use an image data base for scientific iconography (e.g., BioRender), please let us know if you have a license that allows for publication in an academic journal. Often authors use misleading iconography for expedience. Please ensure the information shown is scientifically accurate. If in doubt, please discuss with the editor or provide a sketch so that our designers can create accurate iconography.
- For figures created using a software for editing vector objects like Inkscape, CorelDraw etc., please send the file as a PDF (or SVG, or EPS), PowerPoint or Keynote in which the labels and objects are still editable. For figures created using Adobe Illustrator, please send the Illustrator (.ai) file.

We realize that it is difficult to revise to a specific deadline. In the interest of protecting the conceptual advance provided by the work, we recommend a revision within 3 months (12th Aug 2025). Please discuss the revision progress ahead of this time with the editor if you require more time to complete the revisions.

Referee #1:

Macé et al. have compiled a highly timely review on the structure, assembly and function of bacterial type IV secretion systems. The overall structure of the review is very good, as is the description and the graphical illustration of the structures of the system. The description of the functional mechanisms is somewhat speculative due to a lack of experimental data supporting the hypothesis deduced from the solved structures. The manuscript is good to read over all but falls short in quite a few instances on grammar. I have indicated my corrections and comments in a separate pdf of the manuscript that should be transferred to the authors.

Referee #2:

In their manuscript titled "Type IV Secretion Systems: from structures to mechanisms" Paillard et al. review the research status on Type IV SS in light of recent structural characterization of the R388 type IV SS. Given the complex nomenclature the manuscript starts with a description of the main components of the Type IV SS and then focuses on four main functions: Machinery assembly, pilus biogenesis, donor-recipient cell contact, and substrate recruitment-secretion.

Overall, the manuscript is well written and proposes new data-driven hypothesis to explain the conjugation process. While the manuscript presents original and intriguing hypothesis, the authors should pay more attention to describing what has been structurally demonstrated and what is hypothesized. The authors should also include question marks on the figures (especially fig6) to clearly distinguish hypothetical assemblies from the solved ones.

We think that this review will be beneficial for the community studying the type IV SS and worth publication with minor revisions:

Minor reviews:

Pg 4 "engineered T4S" should be correct to "engineered T4SS"

Pg6 "Finally, VirD4 is an another cytoplasmic ATPase, but with a transmembrane channel" should be correct to "Finally, VirD4 is another" moreover, direct evidence VirD4 transmembrane channel formation is still lacking, and the authors should use a hypothetical period.

Pg10 "Once the IMC is in place, the assembly of the T4SS is finalised when VirB11 binds to the bottom of the IMC, stabilising the double hexameric ATPase complex formed by VirB4 and VirB11": stabilising should be changed to stabilizing. Moreover, VirB4 hexameric ring is stable in absence of Virb11 as shown by cryoET for pKM101 (Pratick Khara, Liqiang Song, Peter J. Christie, Bo Hu 2021) and dot/icm systems (Donghyun Park,David Chetrit,Bo Hu,Craig R. Roy,Jun Liu 2020). In Legionella the Virb11 homolog (DotB) induces a conformational change in the VirB4 homolog DotO, probably activating it.

Pg12 "VirB5 is understood to play" should be changed to "is thought"

Pg12 "VirB5 has structural homology": homology can't be inferred only by structure superimposition - the text should be changed to structural "similarity."

Pg13 "the pilus tip integrates additional proteins (CagI, and CagY)". Indeed, when exposed to epithelial cells H. pylori cells produce two structures: one is a filament and the other a sheathed tube. While CagY is associated with sheathed tubes, is may not form the tip. CagI can be found in association in the OM fraction but direct evidence of its localization at the tip of the pilus is lacking.

Pg20 "Unlike other FtsK family proteins, VirD4 includes an additional transmembrane domain, extending its central channel and creating a full translocation pathway across the inner membrane 66": The authors in reference 66 were able to solve a truncated form of TrwB lacking the transmembrane part, therefore the existence of a full translocation pathway across the inner membrane is yet to be demonstrated. The authors should acknowledge this uncertainty.

Point-By-Point Response

Referee #1:

Macé et al. have compiled a highly timely review on the structure, assembly and function of bacterial type IV secretion systems. The overall structure of the review is very good, as is the description and the graphical illustration of the structures of the system. The description of the functional mechanisms is somewhat speculative due to a lack of experimental data supporting the hypothesis deduced from the solved structures. The manuscript is good to read over all but falls short in quite a few instances on grammar. I have indicated my corrections and comments in a separate pdf of the manuscript that should be transferred to the authors.

General reply to the Referee #1:

We thank the reviewer for their positive assessment of our review and for the valuable feedback. We have carefully addressed all comments and incorporated the suggested corrections from the annotated PDF, thereby improving clarity and grammar throughout. Concerning the more speculative aspects of the functional mechanisms, we have now provided a clearer distinction between experimentally supported findings and hypothetical interpretations, to ensure a more accurate and balanced presentation.

Referee #1 | Comment-1:

“To address these inconsistencies and ensure accurate protein classification, we have compiled a comprehensive reference table, integrating data from UniProt and AlphaFold databases (Table 1 and Supplementary Figure 1).”

I miss the basis of this classification. Alignments, identities, methodology.

Referee #1 | Answer-1:

The classification presented in Table 1 is primarily based on published literature describing known homologues across different T4SSs. In addition, we systematically used AlphaFold structural predictions to validate and, where necessary, correct homology assignments, especially in cases where previous annotations were ambiguous or inconsistent. This integrative approach allowed us to improve and validate the assessment of homology relationships by ensuring better structural consistency. We have now clarified this methodology in the table legend to ensure transparency : “*Protein classification is based on literature-supported homology^{94,95} and refined using AlphaFold structural predictions.*”

Referee #1 | Comment-2:

Does VirB5 assemble at the pilus tip or is it there beforehand and pilus assembly occurs in between B6 and B5 as written above?

Referee #1 | Answer-2:

Thank you for pointing out this ambiguity. The sentence was indeed confusing. VirB5 assembles first, on top of VirB6, forming the initial stalk structure. As the pilus begins to polymerise, it incorporates the pre-assembled VirB5 at its tip, giving rise to the pilus tip complex. We have revised the sentence for clarity as follows: “*The assembly site is tightly surrounded by the Arches, composed of VirB10 and VirB8, suggesting that spatial constraints within this region may facilitate the transition of VirB2-lipid subunits into the pilus assembly. The pilus then grows beneath the VirB5 tip complex, which is pre-assembled on top of VirB6, and pushes upwards to open the initially closed OMCC gate formed by a VirB10 homo-oligomer.*”

Referee #1 | Comment-3:

Also the components IcmS and IcmW were coined chaperones but this term may be a misunderstanding, also for LvgA.

Referee #1 | Answer-3:

We agree that the term “chaperone” may be misleading when referring to IcmS and IcmW. Although these proteins were initially described as chaperones due to their role in facilitating the delivery of certain effectors, recent structural studies of the *Legionella* T4CC have shown that IcmS and IcmW are stably integrated into the core complex. Rather than acting as independent effectors-binding chaperones, they function as constitutive subunits of the T4CC, forming a structural platform that supports substrate recruitment. In contrast, LvgA behaves more like a classical chaperone: it binds specific effectors in the cytoplasm and associates transiently with the T4CC to mediate their recruitment.

It is for that reason that we originally wrote: “*The T4SS_{Dot/icm} of Legionella pneumophila represents the most complex T4CPs identified to date, comprising seven proteins and the chaperone LvgA⁷⁸⁻⁸¹.*” i.e. we used the term “chaperone” exclusively for LvgA and considered IcmS and IcmW as part of the seven core components. However, we have now revised this sentence to clarify this distinction. In the updated version of the manuscript: “*The T4SS_{Dot/icm} of Legionella pneumophila represents the most complex T4CPs identified to date, comprising seven proteins, including IcmS and IcmW as integral subunits of the T4CP, and the chaperone LvgA, which transiently associates with the complex to mediate selective effector recruitment⁷⁸⁻⁸¹*”

Referee #1 | Comment-4:

I think this is really far-fetched. SctRST is quite special. One would need a much more detailed comparison to justify this comparison. I would delete this paragraph and the figure 5b. It is not needed and may lead to wrong assumptions.

Referee #1 | Answer-4:

We agree with the reviewer’s assessment and have removed the paragraph and Figure 5b from the manuscript accordingly.

Referee #1 | Comment-6:

There needs to be a proof of methodology that lead to the table 1. Some seem really far fetched (DotA?). In addition, there are gaps in the table. Does that mean there are less than 12 different components or are the homologs not clear?

Referee #1 | Answer-6:

See also Answer-1 regarding the methodology used to construct Table 1.

Regarding DotA, we agree that this assignment may appear uncertain. We have therefore retained DotA in the table but now marked it with an asterisk. The corresponding note in the legend clarifies the rationale: “* *DotA is an essential inner membrane component of the Dot/Icm system. While lacking sequence homology with VirB6, AlphaFold models suggest structural resemblance. It may act as a functional analogue; IcmF has also been proposed as an alternative⁹⁷.*”

Concerning the gaps in the table, these correspond to cases where homologues could not be confidently identified using existing literature or structural criteria. Rather than speculating, we chose to leave these entries blank to reflect the current level of knowledge and avoid overinterpretation.

Referee #2:

In their manuscript titled "Type IV Secretion Systems: from structures to mechanisms" Paillard et al. review the research status on Type IV SS in light of recent structural characterization of the R388 type IV SS. Given the complex nomenclature the manuscript starts with a description of the main components of the Type IV SS and then focuses on four main functions: Machinery assembly, pilus biogenesis, donor-recipient cell contact, and substrate recruitment-secretion.

Overall, the manuscript is well written and proposes new data-driven hypothesis to explain the conjugation process. While the manuscript presents original and intriguing hypothesis, the authors should pay more attention to describing what has been structurally demonstrated and what is

hypothesized. The authors should also include question marks on the figures (especially fig6) to clearly distinguish hypothetical assemblies from the solved ones.

We think that this review will be beneficial for the community studying the type IV SS and worth publication with minor revisions.

General reply to the Referee #2:

We thank the reviewer for their positive and encouraging feedback. We are glad that the manuscript was found to be both timely and relevant. Following the reviewers' suggestions, we have clarified throughout the text what is structurally demonstrated versus hypothesized. We have also revised the figures and their legends to clearly distinguish hypothetical elements from experimentally resolved components.

Referee #2 | Comment-1:

About "authors should pay more attention to describing what has been structurally demonstrated and what is hypothesized."

Referee #2 | Answer-1:

As suggested by the reviewer, we have carefully revised the manuscript to clearly distinguish between structurally demonstrated features and hypothetical interpretations. We have clarified this distinction throughout the text, and we now explicitly indicate uncertain or modelled elements in the figures by adding question marks to highlight hypothetical assemblies.

Referee #2 | Comment-2:

Pg4 "engineered T4S" should be correct to "engineered T4SS"

Pg6 "Finally, VirD4 is an another cytoplasmic ATPase, but with a transmembrane channel" should be correct to "Finally, VirD4 is another" moreover, direct evidence VirD4 transmembrane channel formation is still lacking, and the authors should use a hypothetical period.

Referee #2 | Answer-2:

Corrected.

Referee #2 | Comment-3:

Pg10 "Once the IMC is in place, the assembly of the T4SS is finalised when VirB11 binds to the bottom of the IMC, stabilising the double hexameric ATPase complex formed by VirB4 and VirB11": stabilising should be changed to stabilizing. Moreover, VirB4 hexameric ring is stable in absence of Virb11 as shown by cryoET for pKM101 (Pratick Khara, Liqiang Song, Peter J. Christie, Bo Hu 2021) and dot/icm systems (Donghyun Park,David Chetrit,Bo Hu,Craig R. Roy,Jun Liu 2020). In Legionella the Virb11 homolog (DotB) induces a conformational change in the VirB4 homolog DotO, probably activating it.

Referee #2 | Answer-3:

Corrected. We have also incorporated the references and findings mentioned by the reviewer to clarify the role of VirB11 and to better reflect the structural observations from pKM101 and *Legionella* Dot/Icm systems.

Referee #2 | Comment-4:

Pg12 "VirB5 is understood to play" should be changed to "is thought"
Pg12 "VirB5 has structural homology": homology can't be inferred only by structure superimposition - the text should be changed to structural "similarity."

Referee #2 | Answer-4:

Corrected as suggested.

Referee #2 | Comment-5:

Pg13 "the pilus tip integrates additional proteins (CagI, and CagY)". Indeed, when exposed to epithelial cells *H. pylori* cells produce two structures: one is a filament and the other a sheathed tube. While CagY is associated with sheathed tubes, it may not form the tip. CagI can be found in association in the OM fraction but direct evidence of its localization at the tip of the pilus is lacking.

Referee #2 | Answer-5:

We agree with the reviewer and have modified the text accordingly to reflect the current uncertainty regarding the exact localisation of CagI and CagY at the pilus tip. *"In more evolved systems, such as Helicobacter pylori T4SS_{cag} system, additional proteins such as CagI and CagY are associated with the surface-exposed structures and contribute to specific interactions with human gastric cells^{9,56,57}. However, their presence at the pilus tip remains to be clearly demonstrated."*

Referee #2 | Comment-6:

Pg20 "Unlike other FtsK family proteins, VirD4 includes an additional transmembrane domain, extending its central channel and creating a full translocation pathway across the inner membrane 66": The authors in reference 66 were able to solve a truncated form of TrwB lacking the transmembrane part, therefore the existence of a full translocation pathway across the inner membrane is yet to be demonstrated. The authors should acknowledge this uncertainty.

Referee #2 | Answer-6:

We acknowledge that the existence of a continuous translocation pathway through VirD4 remains to be demonstrated experimentally. While AlphaFold predicts a transmembrane region in VirD4, the structure cited in reference 66 was solved using a truncated variant lacking this region, and thus cannot confirm such a channel. However, more recently, the structure of VirD4 from *Helicobacter pylori* has been resolved, including the substrate channel, though the outer transmembrane helices remain unresolved. This new data provides partial support for the model; however, we have revised the manuscript to reflect the remaining uncertainty: *"VirD4 is predicted, based on AlphaFold models and partial cryo-EM structures⁷⁰, to include an additional transmembrane domain that may extend its central channel to form a continuous translocation pathway across the inner membrane⁶⁶"*

Conclusion

We sincerely thank the referees for their thoughtful and constructive comments, which greatly contributed to improving the manuscript. This review has been a stimulating effort to synthesise recent advances in the field, and we are grateful for the opportunity to share it with the community. We also thank the editor for the smooth handling of the process, and we are enthusiastic to support the upcoming figure editing stage.

Dear Kevin,

Thank you for submitting the revised version of your review to The EMBO Journal. Your manuscript has now been seen by both original reviewers, who find that their comments have been addressed successfully.

I will now forward your figures to our collaborating graphic editors for final stylistic adaptation. For this purpose, could you please check the spelling and grammar of the text in the figures: Fig. 1d, "contact the pilus" - should it be "contact with the pilus"? Fig. 2c - should it be "cavity CLOSED"? Fig 5b - should it be "Rapid gene expression"? Fig. 6: "donor-recipient cell contact"?

I will also go through the manuscript text myself in the meantime and will let you know if any further textual adjustments to the style might be needed.

Furthermore, there are a few editorial issues that I would need you to address in the final version:

1. Please submit up to five keywords.
2. Please check whether the funding information in our online system and in the manuscript is complete and identical. Currently, the Centre National de la Recherche Scientifique (CNRS) is missing from our online system.
3. Please update references according to The EMBO Journal style - they should be listed in an alphabetical order. Where there are more than 10 authors on a paper, the first 10 should be listed, followed by 'et al.' Please see further information here: <https://www.embopress.org/page/journal/14602075/authorguide#referencesformat>
4. CRediT has replaced the traditional author contributions section because it offers a systematic, machine-readable author contributions format that allows for more effective research assessment. Please remove the Authors Contributions from the manuscript and use the free text boxes beneath each contributing author's name in our online submission system to add specific details on the author's contribution. More information is available in our guide to authors.
5. Please rename "Competing interests" section into "Disclosure and competing interests statement" (further info: <https://www.embopress.org/page/journal/14602075/authorguide#conflictofinterest>).
6. Please provide Table 1 in an editable format.

Please let me know if you have any questions about this final editorial revision. I look forward to working with you to polish up the final version of your manuscript.

With best wishes,

Ieva

Ieva Gailite, PhD
Senior Scientific Editor
The EMBO Journal
Meyerohofstrasse 1
D-69117 Heidelberg
Tel: +4962218891309
i.gailite@embojournal.org

We realize that it is difficult to revise to a specific deadline. In the interest of protecting the conceptual advance provided by the work, we recommend a revision within 3 months (2nd Oct 2025). Please discuss the revision progress ahead of this time with the editor if you require more time to complete the revisions.

Referee #1:

The authors have addressed all points raised by me very well. I have no additional concerns.

Referee #2:

The authors addressed our concerns.

The authors addressed the remaining editorial issues.

Dear Kevin,

Thank you very much for incorporating the final textual changes in your review article. I am now pleased to inform you that your manuscript has been accepted for publication in the EMBO Journal.

For your information, I have attached the final version of the manuscript text below. The blurb that will accompany the title of your article in our online table of contents will be:

"This review summarizes recent structural insights into the assembly and function of bacterial type IV secretion systems."

Your manuscript will be processed for publication by EMBO Press. It will be copy edited and you will receive page proofs prior to publication.

You will soon be contacted by Springer Nature to sign your publishing license. When you login to the customer service website, please use the following token to waive the article publication charges:

MTGZMJU2NTC0NW.

Should you experience any difficulty, please email publishing@embo.org.

If you have any questions, please do not hesitate to contact me or the Editorial Office. Thank you once more for this thought-provoking contribution to The EMBO Journal!

With best wishes,

Ieva
